biotechnology/nanotechnology

*Clinacanthus nutans*, silver nanoparticles, biogenic synthesis, total phenolic content, total flavonoid content

**Author for correspondence:**
Vuanghao Lim
e-mail: vlim@usm.my

This article has been edited by the Royal Society of Chemistry, including the commissioning, peer review process and editorial aspects up to the point of acceptance.

# Optimization of biogenic synthesis of silver nanoparticles from flavonoid-rich *Clinacanthus nutans* leaf and stem aqueous extracts

Siti Nur Aishah Mat Yusuf[1,2],

Che Nurul Azieyan Che Mood[2], Nor Hazwani Ahmad[3], Doblin Sandai[4], Chee Keong Lee[5] and Vuanghao Lim[2]

[1]Department of Chemical Engineering Technology, Faculty of Engineering Technology, Universiti Malaysia Perlis, UniCITI Alam Campus, 02100 Padang Besar, Perlis, Malaysia
[2]Integrative Medicine Cluster, Advanced Medical and Dental Institute, [3]Oncology and Radiological Sciences Cluster, Advanced Medical and Dental Institute, and [4]Infectomics Cluster, Advanced Medical and Dental Institute, Universiti Sains Malaysia, Bertam, 13200 Kepala Batas, Penang, Malaysia
[5]Bioprocess Technology Division, School of Industrial Technology, Universiti Sains Malaysia, 11800 Penang, Malaysia

SNAMY, 0000-0001-9258-5875; NHA, 0000-0001-7353-2495; DS, 0000-0003-0544-4260; CKL, 0000-0003-2668-428X; VL, 0000-0001-5081-0982

*Background*: Silver nanoparticles (AgNPs) are widely used in food industries, biomedical, dentistry, catalysis, diagnostic biological probes and sensors. The use of plant extract for AgNPs synthesis eliminates the process of maintaining cell culture and the process could be scaled up under a non-aseptic environment. The purpose of this study is to determine the classes of phytochemicals, to biosynthesize and characterize the AgNPs using *Clinacanthus nutans* leaf and stem extracts. In this study, AgNPs were synthesized from the aqueous extracts of *C. nutans* leaves and stems through a non-toxic, cost-effective and eco-friendly method. *Results*: The formation of AgNPs was confirmed by UV-Vis spectroscopy, and the size of AgNP-L (leaf) and AgNP-S (stem) were 114.7 and 129.9 nm, respectively. Transmission electron microscopy (TEM) analysis showed spherical nanoparticles with AgNP-L and AgNP-S ranging from 10 to 300 nm and 10 to 180 nm, with average of 101.18 and 75.38 nm, respectively. The zeta potentials of AgNP-L and AgNP-S were recorded at −42.8 and −43.9 mV. X-ray diffraction analysis matched the face-centred cubic structure of silver and was capped with bioactive compounds. Fourier transform

infrared spectrophotometer analysis revealed the presence of few functional groups of phenolic and flavonoid compounds. These functional groups act as reducing agents in AgNPs synthesis. *Conclusion*: These results showed that the biogenically synthesized nanoparticles reduced silver ions to silver nanoparticles in aqueous condition and the AgNPs formed were stable and less toxic.

# 1. Introduction

The field of nanotechnology has undergone substantial developments and has attracted considerable interest from researchers owing to its wide range of application. Generally, nanoparticles include 1–100 nm particles [1]. The physical and chemical properties of nanoparticles are different from those of bulk materials. Nanoparticles have distinct properties because of their high surface-area-to-volume ratios and small sizes [2]. The biological activities and surface energy of nanoparticles increase with their surface areas [3]. Nanoparticles such as silver nanoparticles (AgNPs) are widely used in food industries, biomedicine, dentistry, catalysis, diagnostic biological probes and sensing [4,5].

AgNPs synthesis has become popular, and many studies on the process have been conducted using various chemical, physical and biological methods [6]. Sodium borohydride, hydrazine and ethylene glycol are the most common chemicals used in AgNPs synthesis [7,8]. However, the processes using these chemicals are expensive and harmful to living organisms and hence undesirable. Meanwhile, biological synthesis has attracted interest from researchers because it is cost effective, consumes less energy and is non-toxic [6]. Many studies used natural resources, such as plants [9], bacteria, fungi and yeast to synthesize AgNPs [10]. Nevertheless, the use of plant extracts has become a major interest in AgNPs synthesis. In this method, plant extracts act as reducing and capping agents because of their various bioactive compounds [11,12]. AgNPs synthesis using plant extracts is a preferable option compared to other methods because it eliminates the process of maintaining cell culture and can be scaled up under a non-aseptic environment [13].

The plant extracts that induce AgNP formation contain reducing agents [14], such as the metabolites of phenolic compounds, flavones, alkaloids and sterols [15–17]. Compared with microorganisms, plant extracts are inexpensive, easily available and suitable for industrial use because they do not require purification or the use of cultures [18]. Meanwhile, microbes require aseptic conditions and high maintenance. The time required for the formation of AgNPs is faster for plant extracts compared to microorganisms.

*Clinacanthus nutans* (figure 1) is a small shrub belonging to Acanthaceae [19], and commonly found in tropical Asian countries, mainly Malaysia, Thailand and Indonesia [20]. *C. nutans* is commonly known as 'Belalai gajah' in Malaysia [21], Phaya yo or Phaya plongtong in Thailand and Dandang gendis in Indonesia [20–22]. Several bioactive compounds from *C. nutans* extracts have been isolated and studied. Lupeol, isoorientin, orientin, isovitexin, schaftoside, vitexin and β-sitosterol have been isolated from the stem and leaf extracts of *C. nutans* [23]. Other bioactive groups isolated from this plant were sulfur-containing glucosides and chlorophyll derivatives. Further to this, saponin, phenolics, flavonoids, diterpenes and phytosterols were present in the methanol extracts of the leaves [24]. The aim of this study was to biogenically synthesize the AgNPs from *C. nutans* leaves and stem extracts and to optimize the parameters used in this biogenic synthesis.

# 2. Materials and methods

## 2.1. Plant collection

The dried leaves and stems of *C. nutans* were purchased from Botani Sdn. Bhd, Manjung, Perak. *Clinacanthus nutans* was authenticated with voucher specimen No. 11465 and deposited at Herbarium Unit, School of Biological Sciences, Universiti Sains Malaysia. The dried leaves and stems were pulverized into fine powder with a grinder (Ultra Centrifugal Mill ZM200, Retsch, Haan, Germany). The powder was then sealed in a glass bottle and kept at room temperature until further use.

## 2.2. Plant extraction

The extraction method was adapted from previous study with some modifications [25]. Briefly, 50 g of fine leaf powder of *C. nutans* (CNL) was macerated in 500 ml of distilled water (dH$_2$O) in an orbital shaker

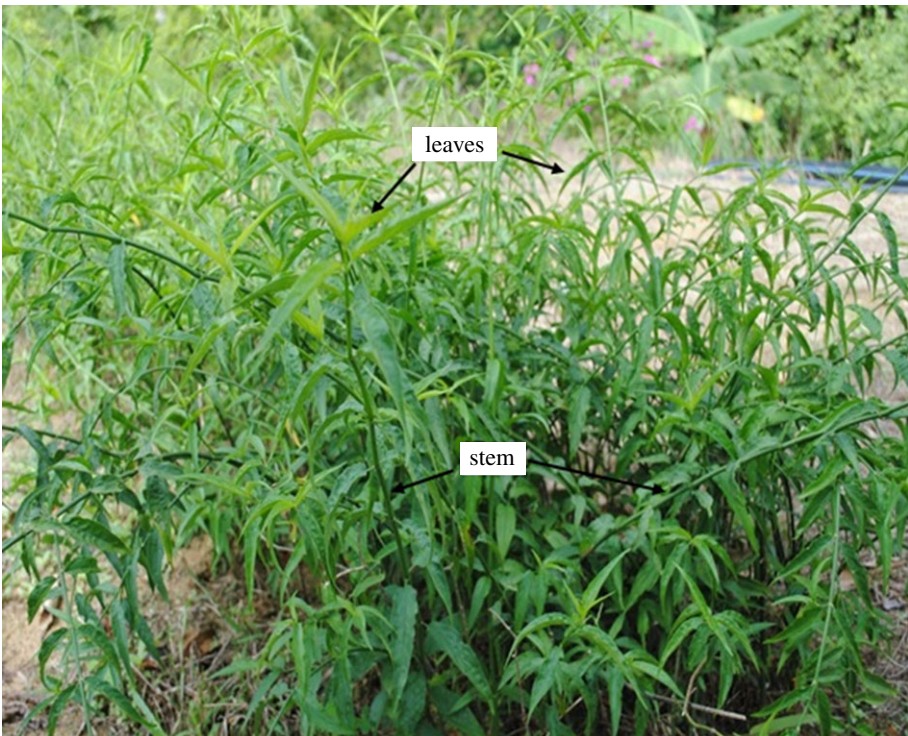

**Figure 1.** *C. nutans* leaves and stems.

(Orbitron, Bottmingen, Switzerland) at room temperature for 24 h at 150 r.p.m. The aqueous extract was then filtered and lyophilized (EYELA FDU 1200, Fisher Scientific, Loughborough, UK). The extract was labelled, weighed and kept in desiccator until further use. The percentage yield of the aqueous extract was calculated according to equation (2.1). These procedures were repeated for powdered stem of *C. nutans* (CNS).

$$\text{Percentage yield} = \frac{w2 - w1}{w0} \times 100\%,\qquad(2.1)$$

where $w0$ is the weight of the initial dried sample, $w1$ is the weight of the container and $w2$ is the weight of the dried extract and container.

## 2.3. Preparation of aqueous extract for qualitative analysis

The qualitative phytochemical analysis was performed for tannins, saponins, alkaloids, proteins, carbohydrates and flavonoids. Each test was expressed as negative (−) or positive (+) reactions. Briefly, 1 g of CNL extract was dissolved in 10 ml of $dH_2O$ [26]. The sample was filtered, and the filtrate was used for further experiments. This procedure was repeated for CNS extract. All screening tests were conducted according to the standard methods [27].

## 2.4. Total phenolic content

The Folin–Ciocalteu method was adapted from previous study with modification [26]. Briefly, the sample of the extract was prepared at concentration of 1 mg ml$^{-1}$ with $dH_2O$. Then, 30 µl of the sample was mixed with 150 µl of 10% (v/v) Folin–Ciocalteu's phenol reagent (diluted with $dH_2O$) in 96-well plate (FalconTM, Fisher Scientific, Loughborough, UK) and the mixture was left for 5 min at room temperature. Then, 120 µl of $Na_2CO_3$ (7.5% w/v) was added and the mixture was incubated for 30 min in the dark. The absorbance was measured at 765 nm with FLUOstar Omega microplate reader (Offenbury, Germany). The amount of total phenolic content (TPC) was calculated in term of µg of gallic acid equivalent (GAE) per mg extract based on the gallic acid standard curve.

## 2.5. Total flavonoid content

Total flavonoid content (TFC) was determined by using aluminium chloride method described by Mishra and co-workers with some modifications [28]. Briefly, the extract was prepared by dissolving

**Table 1.** Parameter variables used to optimize the synthesis of AgNPs.

| parameter | variable |
|---|---|
| extract concentration | 5, 10 and 15% v/v |
| AgNO$_3$ concentration | 1, 3, 5 mM |
| temperature | 25℃, 40℃ and 60℃ |
| time | 15 and 30 min, 1, 2, 4, 8, 10, 12, 20, 22 and 24 h |

1 mg of CNL or CNS in 1 ml of dH$_2$O. Then, 25 µl of each extract was mixed with 125 µl of dH$_2$O and 7.5 µl of NaNO$_3$ (5% w/v). The mixture was incubated for 5 min at room temperature. Next, 15 µl of AlCl$_3$ (10% w/v) solution was added into the mixture and left at room temperature for 6 min before 50 µl of NaOH (1 M) and 27.5 µl of dH$_2$O were added into the mixture. The mixture was incubated in the dark for 20 min. The absorbance was measured at 510 nm using microplate reader. The TFC was expressed as µg quercetin equivalents (QE) per mg extract using quercetin as the standard curve.

## 2.6. Biosynthesis of silver nanoparticles

The synthesis of AgNPs began with the addition of 95 ml of 0.001 M silver nitrate (AgNO$_3$) to 5 ml of CNL extract (1 mg ml$^{-1}$ in dH$_2$O). The mixture was allowed to stir for 24 h. The formation of AgNPs was indicated by the color changes from yellow to dark brown. After 24 h, the mixture was centrifuged (6000 r.p.m., 30 min) and washed with dH$_2$O three times. The pellets were dried overnight in an oven at 40℃ for further characterization [12]. The same procedure was repeated for the CNS extract. Different parameters that affect the synthesis of AgNPs such as extract and AgNO$_3$ concentration, incubation time and temperature were studied. Table 1 shows the parameter conditions used for optimization study.

## 2.7. Characterization of silver nanoparticles

The formation of AgNPs was monitored by measuring the reduction of silver ions with a Perkin Elmer double beam UV-Vis spectroscope (Lambda-25). The sample absorbance was measured between 300 and 600 nm after incubation. The sample was diluted with dH$_2$O at 1 : 1 volume ratio before the measurement [29]. For particle size and zeta potential analysis, the dried powder of AgNPs was suspended in deionized water at 1 : 10 ratio. The measurements were conducted at room temperature with Zetasizer Nano ZS (Malvern, Worcestershire, UK) [12]. For the surface morphology study, the dried AgNPs were placed on the stud and analysed with scanning electron microscope (SEM; Quanta Feg 650) attached with X-Max 50 energy dispersive X-ray spectrometer (EDX; Oxford Instrument, Abingdon, UK). Transmission electron microscope (TEM; Libra 120; Carl Zeiss, Oberkochen, Germany) was used to identify the shapes, sizes and morphologies of the synthesized AgNPs. For TEM analysis, the AgNPs were dispersed in dH$_2$O and sonicated for 15 min. The sample was dropped on a copper grid coated with carbon (400 mesh, 305 mm diameter) and observed [12]. X-ray diffractometer (D8 advance, Bruker, Berlin, Germany) was used to identify the phase and crystallinity of the synthesized AgNPs. All X-ray diffraction (XRD) data were collected in the angular range of $35° \leq 2\theta \leq 80°$ at 40 kV and 30 mA with Cu K$\alpha$ radiation (1.5405 Å). The presence of functional groups that were responsible for the reduction of silver ions was identified with Fourier transform infrared spectrophotometer (FTIR) using KBr disc (Avatar 670; Nicolet, Wisconsin, USA). The absorbance was taken from 4000 to 500 cm$^{-1}$ [29].

# 3. Results and discussions

## 3.1. Plant extraction

The leaf and stem extracts of *C. nutans* were obtained through the maceration method at room temperature. The percentage yield of the leaf and stem extracts are 25.31% and 13.12%, respectively. Maceration method was selected because plants contain thermolabile compounds and are suitable for large scale. Thermolabile compounds degrade or decompose when heated for a long time. In this study, distilled water was chosen as the extraction solvent, which is easily available and able to produce high percentage yield (polar solvent) [30].

**Table 2.** Phytochemical screenings of CNL and CNS.

| phytochemical group | test | crude extract | |
| --- | --- | --- | --- |
| | | CNL | CNS |
| tannins | FeCl$_3$ | + | + |
| saponins | Frothing | + | + |
| alkaloids | Dragendroff's | + | + |
| | Mayer's | + | + |
| amino acid | Ninhydrin | + | + |
| flavonoids | Alkaline reagent | + | + |
| reducing sugar | Fehling | + | + |
| | Benedict | + | + |
| glycosides | Keller–Kilani | + | + |

+ indicates present/positive reaction, − indicates absent/negative reaction.

## 3.2. Qualitative phytochemical analysis

The CNL and CNS were quantitatively analyzed for the presence of bioactive groups. The phytochemical study of CNL and CNS showed that both extracts contain tannins, saponins, alkaloids, amino acid, flavonoids, reducing sugars and glycosides (table 2).

Plant extracts are known to contain large number of phytonutrients. Based on the previous study, saponin, phenolics, flavonoids, diterpenes and phytosterols were present in the methanol extracts of *C. nutans* leaves [31]. The biomacromolecules and secondary metabolites of plant have been widely explored for the *ex vivo* synthesis of metal nanoparticles. According to Mohamad *et al*. [32], the hydroxyl and carbonyl groups of bioactive compounds act as natural reducing agent in formation, capping and stabilizing the nanoparticle synthesis. Therefore, plant extracts are one of the potential sources of natural reducing agents for green synthesis strategies, as this method is easy, efficient, economical and feasible to conduct [33].

## 3.3. Quantitative phytochemical analysis

TPC was obtained by using Folin–Ciocalteau method. The blue colour complex formed from the reduction of phosphotungstic–phosphomolybdic complex by the phenolic compound, and the intensity was measured by microplate reader. The contents of the phenolic compounds were confirmed according to the TPCs of CNL and CNS. The calibration curve of gallic acid ($y = 0.0032x + 0.0407$; $R^2 = 0.998$) was constructed, where the TPC was obtained and expressed as µg of gallic acid equivalents per mg of dry extract (µg GAE/mg dry extract). All the values were expressed as the average ± standard deviation (s.d.). As shown in figure 2, the TPC of CNL was higher (75.52 ± 0.905 µg GAE/mg dry extract) than CNS (37.08 ± 0.477 µg GAE/mg dry extract). TFC was obtained through the aluminium chloride (AlCl$_3$) method. TFC in the samples were calculated by using the calibration curve of quercetin ($y = 0.0003x + 0.0318$; $R^2 = 0.995$) and was expressed as µg of quercetin equivalents per mg of dry extract (µg QE/mg dry extract). All the values were expressed as average ± s.d. Based on the result (figure 2), the TFC of CNL was higher (27.97 ± 1.273 µg QE/mg dry extract) than that of CNS (12.42 ± 1.443 µg QE/mg dry extract).

Based on the results, CNL had higher TPC and TFC compared to CNS. However, in a previous study, the TPC of aqueous leaf extract of *C. nutans* was measured at 30.15 ± 3.09 mg g$^{-1}$ GAE, which was higher than the TPC in this extract [34]. The antioxidant activity may be promoted by the presence of carboxyl group of phenolic which is capable of reducing activity [32]. Phenolic compounds donate hydrogen in nature, hence the presence of phenolic compounds in the plant extract can prevent the oxidative stress-related damage. The variation in phenolic content among the parts of a plant is due to differences in gene expression, which affect the biological properties of the extracts [35,36]. The amount of TPC can vary at the sub-cellular level and within a plant tissue. Moreover, TPC in a plant may vary according to the solvents used, type of extraction methods, age of plant and extraction time [37,38].

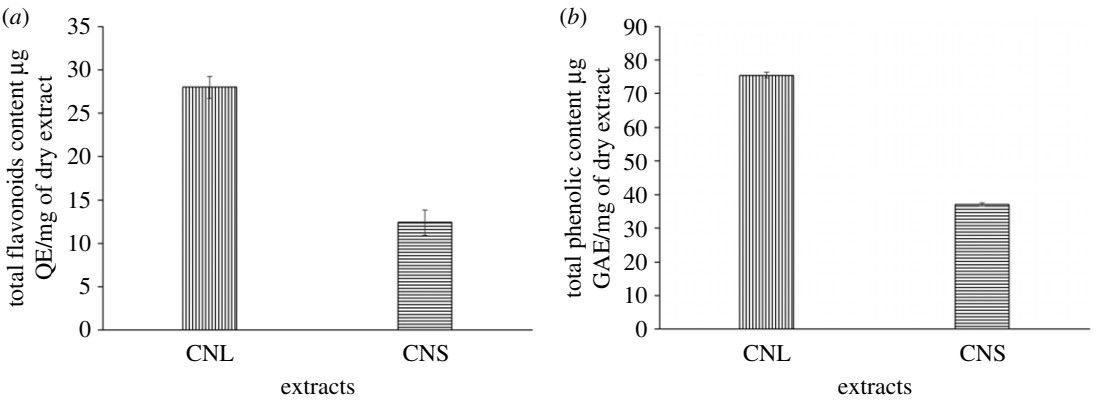

**Figure 2.** (*a*) TFC and (*b*) TPC of CNL and CNS (mean ± s.d., *n* = 3).

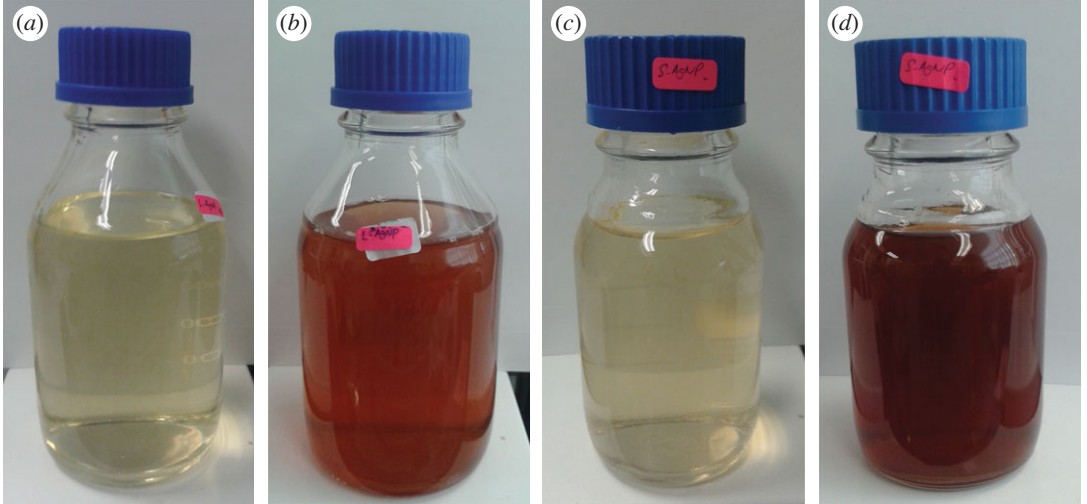

**Figure 3.** Biosynthesis of AgNPs from (*a*) CNL mixed with AgNO$_3$ solution at 0 h, (*b*) after 24 h, (*c*) CNS mixed with AgNO$_3$ solution at 0 h, (*d*) after 24 h.

Apart from phenolic compounds, flavonoids are commonly present in plants and these compounds are beneficial to human health. Previous study reported that the antioxidant activity of flavonoids is due to their hydroxyl groups. Besides, the phytochemical compounds from plant sources can act as reducing and stabilizing agent in AgNPs synthesis [32]. This includes the hydroxyl and carbonyl groups from flavonoids [39]. Therefore, the phytochemicals that were previously identified in *C. nutans* extracts may be responsible for the biosynthesis of AgNPs, where they act as reducing agent in AgNPs synthesis.

## 3.4. Biosynthesis of silver nanoparticles

The reduction of silver ion (Ag$^+$) to AgNPs after the addition of CNL or CNS into the AgNO$_3$ solution was observed after 24 h of incubation. Upon addition of extract into AgNO$_3$ solution, the color of the mixture gradually changed to dark brown (figure 3) indicating the presence of AgNPs due to the conversion of silver ions into AgNPs [40]. Previous study has also reported the same where formation of dark red solution from bright yellow resulted after incubation process [41].

The green synthesis of AgNPs requires the following conditions: (i) types of solvents used for synthesis, (ii) types of reducing agents, and (iii) non-toxic material used for stabilizing the nanoparticles [42]. In this regard, *C. nutans* extracts act as reducing agents which are cheap and easily available without any capping agents. The synthesis of AgNPs using AgNO$_3$ as the precursor and water (solvent) provides high chemical stability and is inexpensive [43]. From the TPC and TFC results, the mechanism of chemical reaction for the formation of AgNPs could be derived from flavonoids and phenolic compounds (reducing agent). These compounds act as electron or hydrogen donors [44]. The ketoform on the backbone of a flavonoid compound reduces from Ag$^+$ to Ag$^0$ [45].

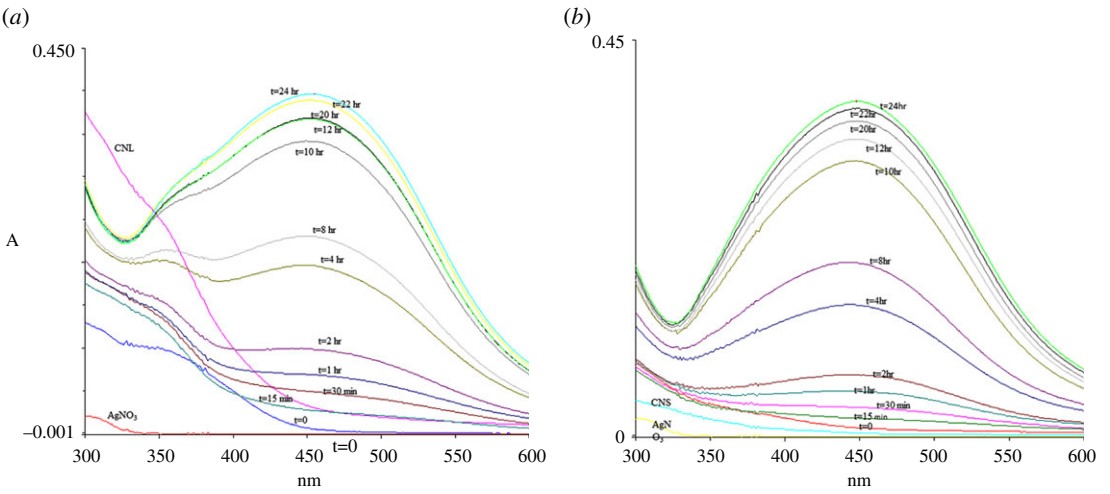

**Figure 4.** UV-Vis spectra of (*a*) AgNP-L and (*b*) AgNP-S as a function of time. Reaction of 5% *C. nutans* extract in 1 mM AgNO₃ at 25°C.

FTIR analysis was conducted to confirm the presence of the functional groups present in the extract that may be responsible in the formation of AgNPs.

## 3.5. UV-Vis spectroscopy analysis

UV-Vis spectroscopy is an important technique to observe the formation of AgNPs by monitoring the optical properties and electronic structure of the synthesized nanoparticles. The electron cloud of a nanoparticle can oscillate on the surface of the nanoparticle, which absorbs electromagnetic waves at a particular frequency. This phenomenon is called surface plasmon resonance (SPR) and recorded as electromagnetic wavelengths by UV-Vis spectrophotometer [46]. Figure 4 shows the optimization of incubation time during AgNPs synthesis. Based on the spectra, UV wavelengths of AgNP-L (AgNPs from leaf extract) and AgNP-S (AgNPs from stem extract) were taken from 15 min until 24 h of incubation. The peaks intensified as the incubation time increases. The increase in intensity of the wavelength absorbance is due to the increasing number of nanoparticles formed from the reduction of silver ions and biomolecules in the aqueous plant extract solution [47,48]. From the result, 24 h was selected as the best incubation time for AgNPs synthesis. Besides that, figure 4 shows the wavelength absorbance of both leaves and stem extracts. The higher absorbance intensity from leaves extract showed a higher flavonoid and its derivatives content compared to stem extract [49]. This correlates with the results from TPC which showed leaves extract has 27.97 ± 1.273 µg QE/mg of dry extract compared to stem extract at 12.42 ± 1.443 µg QE/mg of dry extract.

The parameters of extract concentrations, AgNO₃ concentrations and temperatures were conducted to optimize the AgNPs synthesis. The analysis for extract concentration was performed at 25°C for 24 h with extract concentrations at 5%, 10% and 15% (v/v). The synthesized AgNP-L peaks were observed between 440 and 454 nm, whereas 441–447 nm for AgNP-S (figure 5). As the concentrations of CNL and CNS increase, the sharpness of the absorption peaks was also increased. The sharpness of the peak indicates the formation of spherical shape of synthesized AgNPs and homogeneous distribution [50]. Moreover, SPR shifts were also observed along the increasing concentration of extract concentration for both leaves and stem synthesis. These shifts showed reduction in size of the synthesized AgNPs [50]. Based on the results, 15% (v/v) of both extracts were selected for the optimization of next parameter.

The effect of AgNO₃ solution was then studied. Different concentrations of AgNO₃ (1, 3 and 5 mM) were tested with 15% (v/v) of *C. nutans* extract at 25°C and 24 h incubation time. The absorption peaks of the synthesized AgNP-L and AgNP-S were found between 442–455 nm and 445–450 nm, respectively (figure 6). The highest concentration of AgNO₃ yielded the highest absorption with slightly lower wavelength which indicates the highest amount of AgNPs being synthesized. Therefore, based on the absorption spectrum, 5 mM AgNO₃ showed the highest formation of AgNPs for both extracts. Thus, this concentration was used for subsequent experiments [51].

Finally, the effect of temperature on the synthesis of AgNPs was also studied. Different temperatures tested, i.e. 25°C, 40°C and 60°C with 15% (v/v) of *C. nutans* extracts and 5 mM of AgNO₃. The reaction mixture was incubated for 24 h. The absorbances of AgNP-L and AgNP-S increased with temperature, with the highest absorbance obtained at 60°C (figure 7). Previous study on the synthesis of AgNPs

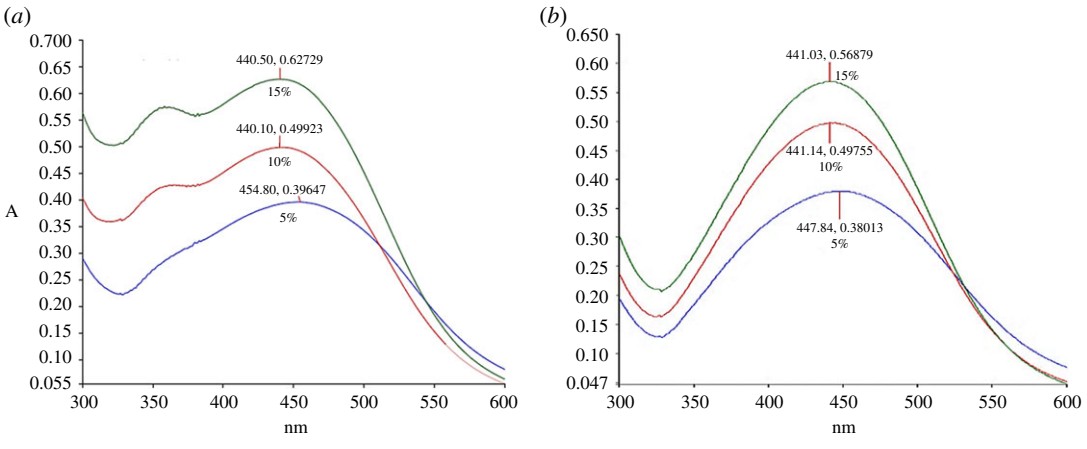

**Figure 5.** UV–Vis spectra of (*a*) AgNP-L and (*b*) AgNP-S for different concentrations of *C. nutans* extract.

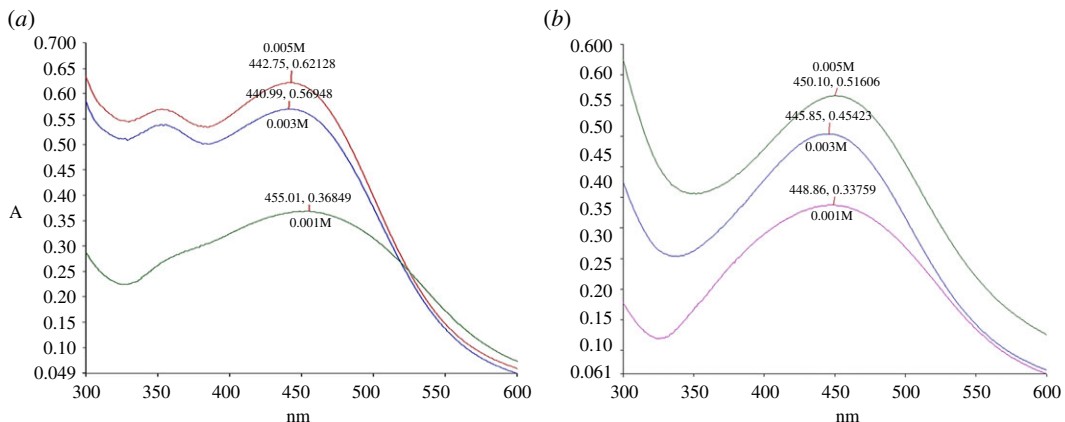

**Figure 6.** UV–Vis spectrum of (*a*) AgNP-L and (*b*) AgNP-S at different concentrations of $AgNO_3$.

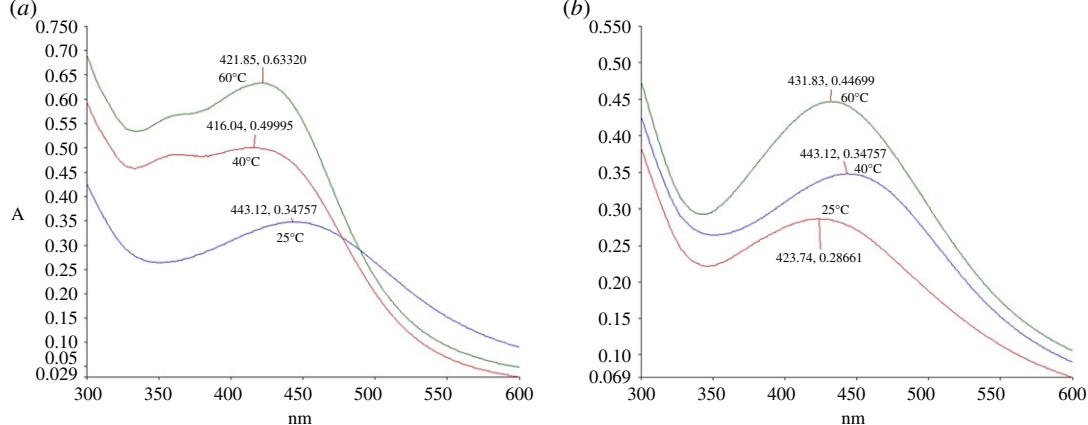

**Figure 7.** UV–Vis spectra of (*a*) AgNP-L and (*b*) AgNP-S at different temperatures.

using *Pulicaria glutinosa* extract showed that AgNPs production increased proportionally with temperature [29]. The maximum absorption peaks for both synthesized AgNP-L and AgNP-S were between 421–443 and 423–443 nm, respectively. The temperature of 60°C was selected for both AgNP-L and AgNP-S as the amount of AgNPs synthesized is highest with average mean diameter of the nanoparticles.

The peak for 60°C is narrower compared to the peaks at 25°C and 40°C. According to Khan *et al.* [29], the absorbance peak is broad at a high wavelength when the particle size of nanoparticles increases. A narrow peak at short wavelength indicates small particle size. From these analyses, it was

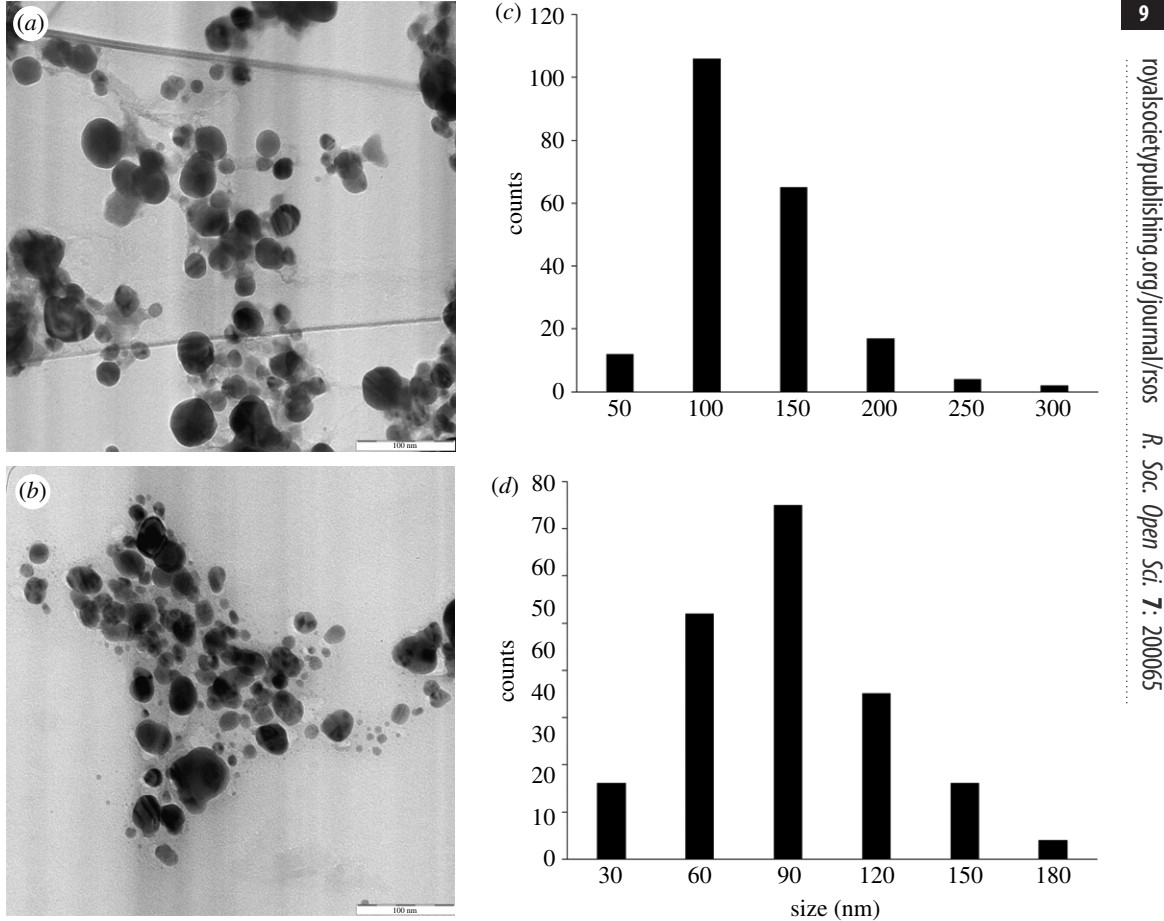

**Figure 8.** TEM images of (*a*) AgNP-L, (*b*) AgNP-S with magnifications of 63 000 × and size distribution of (*c*) AgNP-L and (*d*) AgNP-S.

concluded that the optimum conditions for the synthesis of AgNPs were 15% v/v of plant extract in 5 mM of AgNO$_3$ solution at the temperature of 60°C and 24 h incubation time.

The SPR bands became sharper and shifted to shorter wavelengths when the temperature increases, indicating decrease in particle size. The reason for the decrease is the increase in reduction rate during the synthesis process. Silver ions are consumed during the process, thus blocking any secondary reduction process on the surface of the AgNPs [24]. The shape of the synthesized particles can be predicted theoretically, with spherical shape of particles appearing at the wavelengths within the range of 400–490 nm, pentagon (500–590 nm) and triangular (600–700 nm) [50,52]. From the UV-Vis spectra, the morphology of the synthesized AgNPs were mostly spherical as the maximum absorption occurred at 421 nm (AgNP-L) or 431 nm (AgNP-S).

## 3.6. Particles size and zeta potential measurement

The size of the synthesized AgNPs was determined through dynamic light scattering (DLS) and transmission electron microscopy (TEM). In DLS measurement, the average size distribution was recorded at 114.7 ± 1.012 nm for AgNP-L and 129.9 ± 1.400 nm for AgNP-S. The polydispersity indices (PDIs) for AgNP-L and AgNP-S were 0.206 ± 0.007 and 0.183 ± 0.004, respectively (table 3).

The TEM images showed that AgNP-L was spherical in shape and had a size distribution ranging from 10 to 300 nm (average = 101.18 nm; figure 8). AgNP-S was also spherical in shape and had a size distribution range of 10–180 nm (average = 75.38 nm).

The DLS measures the average size of nanoparticles in liquid suspension, which requires low volume of sample. This measurement applies the Brownian motion theory for particle size. Brownian motion is a random movement of particles in a gas or suspension. The average size of the nanoparticles was determined by measuring the dynamic fluctuation from the light scattering intensity and velocity movement of the particles in suspension [53]. The calculated PDI values of the samples were

**Table 3.** Particle size and zeta potential analysis.

| samples | polydispersity index (PDI) | particle size (nm) | zeta potential (mV) |
|---------|---------------------------|--------------------|--------------------|
| AgNP-L | 0.206 ± 0.007 | 114.7 ± 1.012 | −42.8 |
| AgNP-S | 0.183 ± 0.004 | 129.9 ± 1.400 | −43.6 |

0.206 ± 0.007 (AgNP-L) and 0.183 ± 0.004 (AgNP-S), which are within the range of 0–1, where 0 refers to monodisperse and 1 is polydisperse [53]. This result clearly shows that the synthesized AgNPs were in a monodisperse phase, and the particle aggregation is minimal. The size, morphology and stability of the synthesized AgNPs are influenced by the experimental conditions, such as reducing and stabilizing agents in synthesis process [6]. Saxena *et al*. [54] and Shameli *et al*. [55] suggested that phenolic and other compounds present in the plant extract caused the agglomeration of the nanoparticles. However, the average size of synthesized AgNPs in DLS measurements are larger than those in TEM analysis because DLS measures the nanoparticles in clusters [56].

Zeta potential calculates the net charges of a moving particle under the electric field in solution [57]. Negative zeta potential was observed in AgNP-L and AgNP-S (table 3). The negative value represents the net charges around a particle and not the actual surface charge. The negative charge may be due to the adsorption of bioactive compounds, such as polyphenolic compounds onto the AgNPs surface [58]. Temperature plays an important factor in the stability of the nanoparticles and increases with zeta potential. The stability of the AgNPs can be determined by the zeta potential values. High zeta potential causes strong repulsive force among the particles, thus prevent them from aggregating [4].

## 3.7. Scanning electron microscopy–energy dispersive X-ray analysis

The surface morphology and elemental compositions of the synthesized AgNPs were determined by scanning electron microscopy–energy dispersive X-ray (SEM-EDX). Based on the SEM images, the AgNP-L and AgNP-S had rough surfaces and were spherical in shape. The EDX spectra showed intense peak at 3 keV, suggesting that Ag is the main element in AgNPs (figure 9). The same results were reported by Li *et al*. [59], where peak at 3 keV showed weak carbon, oxygen and chlorine. This is due to the binding of biomolecules from the plant extract onto the surface of AgNPs [29,60].

## 3.8. X-ray diffraction analysis

XRD was used to determine the crystalline phase of the synthesized AgNPs. The XRD spectra of AgNPs obtained ranged from 20° to 80°. The peaks at $2\theta$ values for AgNP-L were 38.42° (111), 46.43° (200), 65.31° (220) and 76.89° (311). Figure 10 shows the peaks for AgNP-S at 38.32° (111), 46.35° (200), 64.56° (220) and 76.85° (311). The unassigned peaks with (*) symbol were recorded at 27.94°, 32.31°, 54.97° and 67.74° (AgNP-L); 27.94°, 32.33° and 54.89° (AgNP-S).

The observed peaks were compared with the silver database from the Joint Committee on Powder Diffraction Standard (JCPDS) or International Centre for Diffraction Data (ICDD). Four peaks were indexed as 111, 200, 220 and 311 on the synthesized AgNPs and they matched the face-centred cubic (FCC) structure of silver (JCPDS file no. 04-0783) [61]. The positions of the peaks were slightly shifted because of the presence of strain in the crystal structure as part of the characteristic of nanocrystalline [62].

The unassigned peaks with (*) symbol recorded at 27.94°, 32.31° and 54.97° for AgNP-L, whereas 27.94°, 32.33° and 54.89° for AgNP-S, were similar to the reported unassigned peaks in studies that used cow's milk, palm oil mill effluent (POME), *Annona squamosa* peel and *Murraya koenigii* extracts [48,63–65]. In addition, the peaks recorded at 32.31° and 54.97° were similar to those reported as silver oxide [66]. The peaks with (*) are related to organic compound in plant extract [67,68].

## 3.9. Fourier transform infrared spectrophotometer spectroscopy analysis

FTIR was conducted to determine the possible functional groups responsible for the reduction of Ag⁺. The comparison was made between *C. nutans* extract and synthesized AgNPs at wavenumbers from 4000 to 500 cm⁻¹. The FTIR spectra in figure 11 show the peaks at 3415.97, 1623.42, 1408.57 and 1093.41 cm⁻¹ in CNL and 3415.08, 1616.26, 1410.77 and 1043.45 cm⁻¹ in CNS. AgNP-L exhibited peaks at 3414.10, 1622.25 and 1382.47 cm⁻¹ while 3416.83, 1624.68 and 1383.51 cm⁻¹ for AgNP-S.

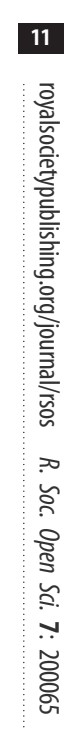

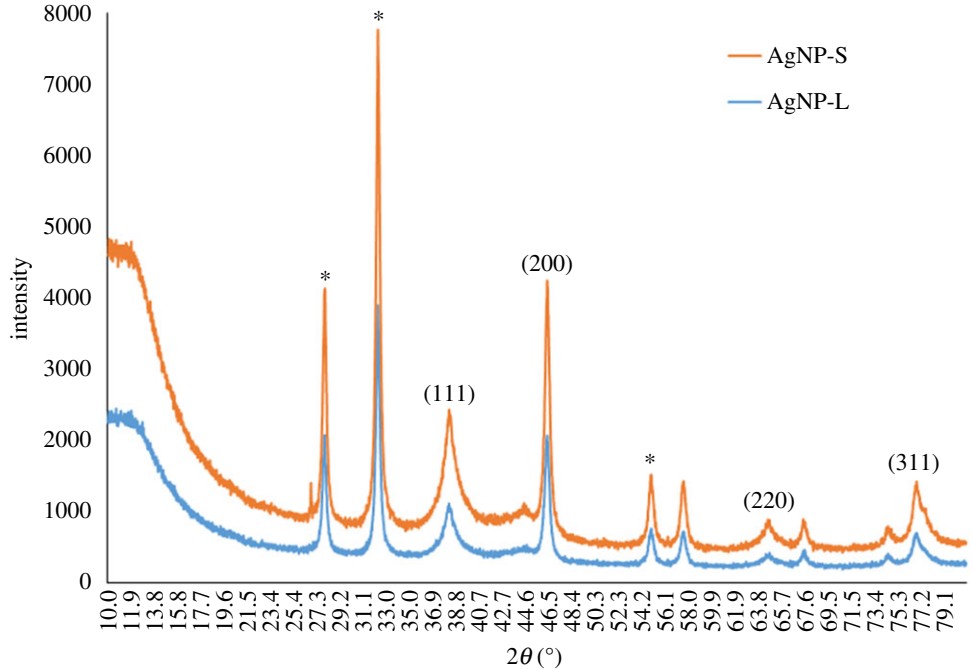

**Figure 9.** SEM images of (*a*) AgNP-L, (*b*) AgNP-S with magnifications of 100 000 × and EDX spectra of (*c*) AgNP-L, (*d*) AgNP-S.

**Figure 10.** XRD spectra of AgNP-L and AgNP-S.

The peaks at 3415.97 and 3415.08 cm$^{-1}$ for CNL and CNS are represented by the -OH or -NH group [55]. Based on phytochemical analysis, both CNS and CNL extracts contain phenolic compounds that might be involved in the reduction and formation of AgNPs [69]. The -OH group in the *C. nutans*

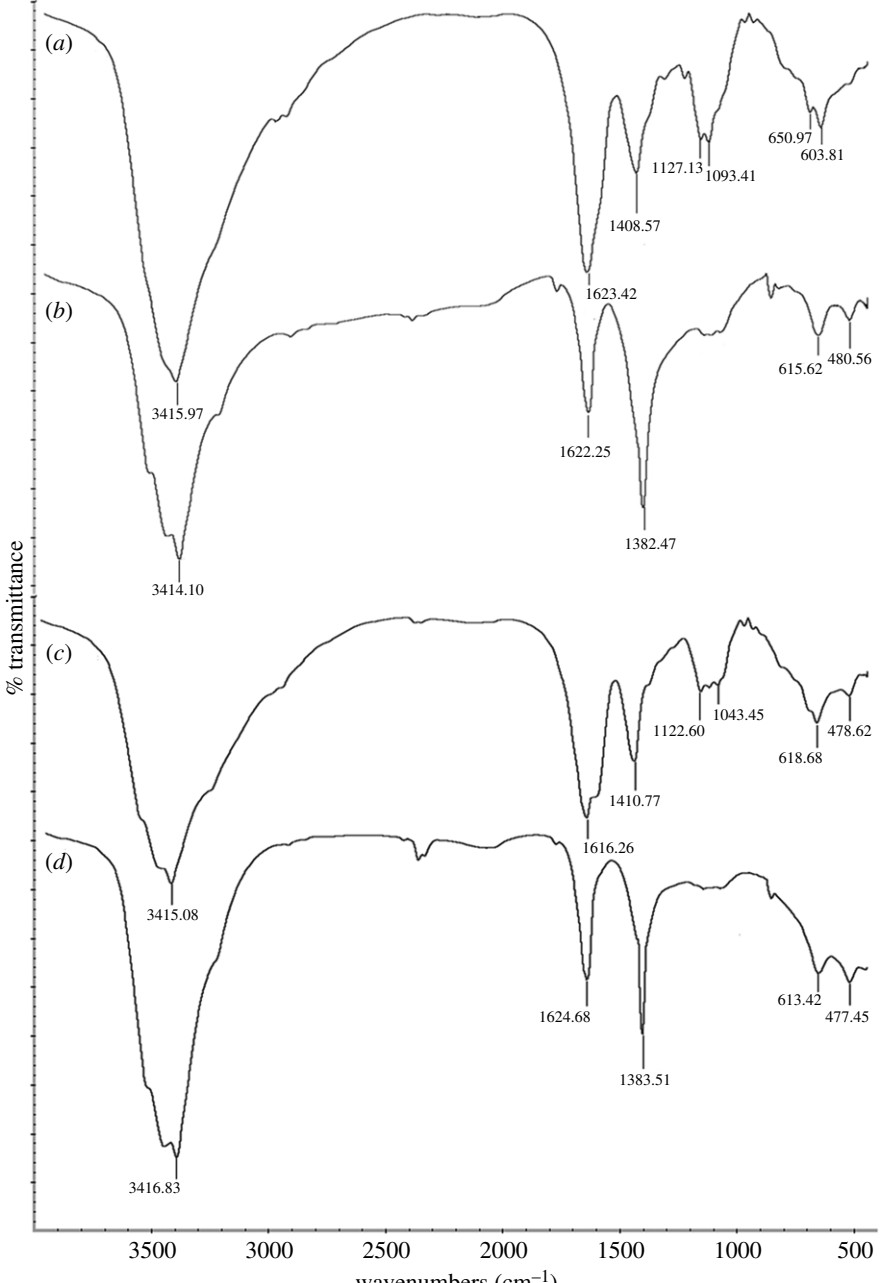

**Figure 11.** FTIR spectra of (*a*) CNL, (*b*) AgNP-L, (*c*) CNS and (*d*) AgNP-S.

extract may be responsible for the oxidation of alcohol to aldehyde during the reduction of $Ag^+$ to $Ag^0$ [70]. The peaks at 1623.42 cm$^{-1}$ (CNL) and 1616.26 cm$^{-1}$ (CNS) shifted to 1622.25 cm$^{-1}$ (AgNP-L) and 1624.68 cm$^{-1}$ (AgNP-S) because of the carbonyl stretch in amide [71]. These peaks are related to the binding of AgNPs to amide groups that enhance the stability of AgNPs [72,73]. Ajitha *et al.* [74] also suggested that the proteins of plant extracts may act as surfactant to stabilize AgNPs. The peaks at 1408.57 cm$^{-1}$ (CNL) and 1410.77 cm$^{-1}$ (CNS) for both plant extracts shifted to 1382.47 cm$^{-1}$ (AgNP-L) and 1383.51 cm$^{-1}$ (AgNP-S). This corresponds to the C-O-H bending (in-plane) of carboxylic acids vibrations [75]. The peaks of both *C. nutans* extracts at 1093.41 and 1043.45 cm$^{-1}$ show the major reductions on the AgNPs FTIR spectra. The peaks also indicate the presence of C-N of aliphatic amines [76].

*C. nutans* aqueous extract has been reported to contain various bioactive compounds, including phenolics, flavonoids, stigmasterol, β-sitosterol and chlorophyll derivatives [77]. These compounds were hypothesized to be responsible for the formation of AgNPs. *C. nutans* has been reported to contain high percentage of phenol and flavonoids. High-performance liquid chromatography analysis

conducted by Sarega *et al*. [78] on the aqueous leaf extract of *C. nutans* showed the presence of four major compounds, namely, protocatechuic acid, chlorogenic acid, caffeic acid and ferulic acid. Although there is no clear evidence on which specific bioactive compound is responsible for the formation AgNPs, it is highly accepted that these phytochemical compounds help in the formation of AgNPs. Phenolics and flavonoids in the plant extract may be responsible for the bioreduction of $Ag^+$ to AgNPs.

# 4. Conclusion

The biogenic synthesis of AgNPs from *C. nutans* leaf and stem aqueous extracts was performed. The process was found to be economical, non-toxic and environmentally friendly. The synthesized AgNP-L and AgNP-S were spherical in shape and had average sizes of 114.7 and 129.9 nm, respectively. Tannins, saponins, alkaloids, amino acid, reducing sugar, flavonoids and glycosides were detected from the extracts. The optimized parameters for *C. nutans* leaf and stem aqueous extract-mediated synthesized silver nanoparticles were obtained as follows: 15% (v/v) of extract, 15 mM of $AgNO_3$, 60°C and 24 h incubation time. These parameters affect the shape of the nanoparticles, polydispersity index and particle size distribution. The FTIR analysis showed the presence of functional groups such as -OH, -NH, -C=O and -COOH, which suggests the conjugation of phenolics and flavonoids with silver ions in the plant extracts.

Data accessibility. Lim, Vuanghao *et al*. (2020), Optimization of biogenic synthesis of silver nanoparticles from flavonoid-rich *Clinacanthus nutans* leaf and stem aqueous extracts, Dryad, Dataset, https://doi.org/10.5061/dryad.905qfttgr [79].

Authors' contributions. V.L. designed the study; S.N.A.M.Y. and C.N.A.C.M. conceived the study. Data were collected by S.N.A.M.Y. and C.N.A.C.M. with help from N.H.A. and D.S. C.K.L. helped with characterization and result analysis. This manuscript was prepared by S.N.A.M.Y. and C.N.A.C.M., and all authors contributed to revisions and approved the final manuscript.

Competing interests. The authors declare that they have no competing interests.

Funding. The authors would like to thank Universiti Sains Malaysia for funding support from Bridging Grant (304.CIPPT.6316264).

Acknowledgements. The authors would like to thank Integrative Medical Cluster Laboratory Management for research permission and facilitation. Special thanks for Masturah Narawi, Asila Dinie Ayub, Hock Ing Chiu, Zaleha Md Toha and Siti Fatimah Samsurrijal for their assistance with the laboratory and technical work.

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
