## [Reviewer comments · Royal Society Open Science]

Review History

RSOS-200065.R0 (Original submission)

Review form: Reviewer 1

Is the manuscript scientifically sound in its present form?

Yes

Are the interpretations and conclusions justified by the results?

Yes

Is the language acceptable?

Yes

Do you have any ethical concerns with this paper?

No

Have you any concerns about statistical analyses in this paper?

No

Recommendation?

Accept with minor revision (please list in comments)

Comments to the Author(s)

The work of Lim, Vuanghao et al. reports a new sustainable method to obtain Ag nanoparticles. The presented approach involved the application of *Clicanthus nutans* aqueous extract of two different plant sections and studied the impact of time, temperature and solutions concentration. A complete and interesting characterization both of extracts and particles was performed. It is opinion of this referee that the work is worthy of publication on Royal Society Open Science. In order to better fit inside Journal requirements and completely fulfill the scientific purpose, some minor modifications/additions should be considered.

Comments are presented hereafter, divided according to pages and lines.

Pag3

Line19: The reported NP dimensions should be expressed according to the applied techniques (in this case DLS) and with the proper form (i.e. 114 nm should be 114.7 ecc...). Furthermore, as correctly said in the text (Pag 12 line 4/5), DLS is not able to discriminate clusters from NPs. A more representative information could be offered reporting the TEM NPs average size (AgNP-L: 101.18 nm, AgNP-S: 75.38 nm).

Pag4

Line8 and 10 too many repetitions of *C. nutans*.

Pag5:

Line 35/36: is the L/L ratio (95/5) been optimized before?

Line38: Do the authors expect AgNPs to be stable at that temperature under air? Oxidations can be ignored?

Line54-55: It is not clear which part of the sample was analysed.

Pag6

Line47-50: The opinion of this referee is that a better contextualization of the state of the art should be necessary to valorise the work presented.

Pag7

Line25: For the sake of comparison, it should be better to express concentrations with the same dimensions (micrograms or milligrams).

Line34-35: It is not completely correct to assign the flavonoids antioxidant activity simply to hydroxyl groups, considering that rings and insaturations are involved in the radical polyphenol stabilization, once the molecule scavenged the oxidation source. Furthermore, even phenolic acids such as compounds belonging to the families of hydroxycinnamic acids and hydroxybenzoic acids should be cited in the text, considering the role displayed by carboxyl groups, according to the evidences reported in the work (i.e. Pag14 line 60 but above all Pag15 line 10/11).

Line 41: Are the solutions always exposed to sunlight? Silver is sensible to light, and the irradiation can play a critical role in the NPs formation. It is opinion of this referee that a comparison test should be conducted in dark conditions, with all the other optimized parameters unchanged (CNL and CNS).

Pag8

Line40: Please, modify magnetic in electromagnetic.

Pag9

General UV-vis: Authors did not make any consideration about the presence of a shoulder for AgNP-L system. This peak seems to be peculiar of the leaves extract (as the CNL "blank" states), not appearing for AgNP-S, suggesting a different nature between the two phytochemicals.

Some considerations should be necessarily given to the reader, being crucial for the purpose depicted by the presented paper. Furthermore a dedicated sentence should be inserted in order to exclude the possibility that this shoulder at lower nm can indicate another populations of particles size.

Line24/25: Do the authors have any hypothesis about nm shifts, between AgNP-L and AgNP-S peaks? Usually in literature UV-Vis bathochromic or hypsochromic shifts are correlated to different NPs dimensions. This consideration can fit in the next size evaluation made by TEM, unveiling different NPs dimensions.

Line28: This statement should contain a correlated reference (see comment at Pag10 Line58)

Figure 5: The AgNPs obtained with 5% extract concentration are, both for L and S, shifted to higher wavelengths. This phenomenon depicts a different size in the material. The opinion of this reviewer is that a short comment should be integrated in the text.

Pag10

Figure 6: Similarly to Figure 5, the same observation can be done for more diluted solutions of AgNO₃ (0.001M).

Line46/50: Similarly to Figure 5 and 6, the same observation can be done for lower T (25°C).

Line58: The reference "50" do not express data about the NPs shape assignation according to registered wavelength. Probably the original insertion was Ref 49, that in any case just cites the useful paper: Optical Properties of Metal Nanoparticles with Arbitrary Shapes. J. Phys. Chem. B 2003, 107, 26, 6269-6275. Please add the correct reference.

Pag13

Line6: The symbol * is not properly reported inside the figure.

Line46: For the sake of clarity, authors should briefly report in the text how they exclude the formation of any traces of silver oxides. This information can be easily extrapolated from the XRD, and is opinion of this referee that this point should be encountered, because of the numerous literature examples of oxides synthesized with similar approaches. A related sentence will enrich and valorise both the novelty and the efficacy of the presented work.

Pag15

Line21: Similarly to Pag 3 line19, the reported NP dimensions should be expressed according to the applied techniques and with the proper. Furthermore, 119 probably is a typo of 129.9 nm.

Review form: Reviewer 2

Is the manuscript scientifically sound in its present form?

No

Are the interpretations and conclusions justified by the results?

No

Is the language acceptable?

Yes

Do you have any ethical concerns with this paper?

No

Have you any concerns about statistical analyses in this paper?

No

Recommendation?

Reject

Comments to the Author(s)

Authors tried to prepare silver nanoparticle with help of natural herb of leaf and stem extract. There are several studies are reported in this area. No novelty in this study.

Review form: Reviewer 3**Is the manuscript scientifically sound in its present form?**

Yes

Are the interpretations and conclusions justified by the results?

Yes

Is the language acceptable?

No

Do you have any ethical concerns with this paper?

No

Have you any concerns about statistical analyses in this paper?

No

Recommendation?

Major revision is needed (please make suggestions in comments)

Comments to the Author(s)

Major review.

In this manuscript, the authors synthesized AgNPs using the aqueous extracts of *C. nutans* leaves and stems through a non-toxic, cost effective and eco-friendly method and determined the classes of phytochemicals, to biosynthesis and characterise the AgNPs. The structure of the manuscript is clear, but there are some grammatical problems in details and some confusion in the selection of the optimal conditions for synthesis. Therefore, I recommend this manuscript to be accepted after solving the following issues.

1. Some sentences in the manuscript have grammatical errors, the sentence in line 40 page 2 should be "AgNP synthesis has become popular", the sentence in line 47 page 2 should be "has become a major interest". The tenses need to be unified in line 8,9,10 page 3. What's more, the use of phrase "AgNPs synthesis" should be uniform throughout the manuscript.
2. There is no reference to this sentence "The biological activities and surface energy of nanoparticles increase with their surface areas." in the first paragraph in Introduction.
3. You mentioned that the use of plant extracts had become a major interest in AgNPs synthesis, what's new about your method compared to other methods using plant extracts to synthesis AgNPs? You may give comparative experiments or the novelty of your method to explain this.
4. The pictures in figure 1 should be reselected for these pictures are not suitable enough for publication.
5. The variable-controlling approach using part 4.5 is inappropriate. The interference of multiple conditions on the result may be ignored if the optimal conditions of a certain factor is selected as the premise in the selection of optimal conditions of next factor. It is suggested to use the orthogonal experiment method, so it is necessary to give the mathematical thought of designing the orthogonal experiments and add supplementary experiments.
6. The X-coordinate and Y-coordinate in UV-VIS spectra in figure 4,5,6,7 are not clear enough.

Decision letter (RSOS-200065.R0)

Dear Dr Llm:

Title: Optimisation of biogenic synthesis of silver nanoparticles from flavonoid-rich *Clinacanthus nutans* leaf and stem aqueous extracts
Manuscript ID: RSOS-200065

The editor assigned to your manuscript has now received comments from reviewers. We would like you to revise your paper in accordance with the referee and Subject Editor suggestions which can be found below (not including confidential reports to the Editor). Please note this decision does not guarantee eventual acceptance.

Please submit your revised paper before 16-May-2020. Please note that the revision deadline will expire at 00.00am on this date. If we do not hear from you within this time then it will be assumed that the paper has been withdrawn. In exceptional circumstances, extensions may be possible if agreed with the Editorial Office in advance. We do not allow multiple rounds of revision so we urge you to make every effort to fully address all of the comments at this stage. If deemed necessary by the Editors, your manuscript will be sent back to one or more of the original reviewers for assessment. If the original reviewers are not available we may invite new reviewers.

RSC Associate Editor:
Comments to the Author:
(There are no comments.)

RSC Subject Editor:
Comments to the Author:
(There are no comments.)

Reviewers' Comments to Author:
Reviewer: 1

Comments to the Author(s)
The work of Lim, Vuanghao et al. reports a new sustainable method to obtain Ag nanoparticles. The presented approach involved the application of *Clicanthus nutans* aqueous extract of two different plant sections and studied the impact of time, temperature and solutions concentration. A complete and interesting characterization both of extracts and particles was performed. It is opinion of this referee that the work is worthy of publication on Royal Society Open Science. In order to better fit inside Journal requirements and completely fulfill the scientific purpose, some minor modifications/additions should be considered.

Comments are presented hereafter, divided according to pages and lines.

Pag3

Line19: The reported NP dimensions should be expressed according to the applied techniques (in this case DLS) and with the proper form (i.e. 114 nm should be 114.7 ecc...). Furthermore, as correctly said in the text (Pag 12 line 4/5), DLS is not able to discriminate clusters from NPs. A more representative information could be offered reporting the TEM NPs average size (AgNP-L: 101.18 nm, AgNP-S: 75.38 nm).

Pag4

Line8 and 10 too many repetitions of *C. nutans*.

Pag5:

Line 35/36: is the L/L ratio (95/5) been optimized before?

Line38: Do the authors expect AgNPs to be stable at that temperature under air? Oxidations can be ignored?

Line54-55: It is not clear which part of the sample was analysed.

Pag6

Line47-50: The opinion of this referee is that a better contextualization of the state of the art should be necessary to valorise the work presented.

Pag7

Line25: For the sake of comparison, it should be better to express concentrations with the same dimensions (micrograms or milligrams).

Line34-35: It is not completely correct to assign the flavonoids antioxidant activity simply to hydroxyl groups, considering that rings and insaturations are involved in the radical polyphenol stabilization, once the molecule scavenged the oxidation source. Furthermore, even phenolic

acids such as compounds belonging to the families of hydroxycinnamic acids and hydroxybenzoic acids should be cited in the text, considering the role displayed by carboxyl groups, according to the evidences reported in the work (i.e. Pag14 line 60 but above all Pag15 line 10/11).

Line 41: Are the solutions always exposed to sunlight? Silver is sensible to light, and the irradiation can play a critical role in the NPs formation. It is opinion of this referee that a comparison test should be conducted in dark conditions, with all the other optimized parameters unchanged (CNL and CNS).

Pag8

Line40: Please, modify magnetic in electromagnetic.

Pag9

General UV-vis: Authors did not make any consideration about the presence of a shoulder for AgNP-L system. This peak seems to be peculiar of the leaves extract (as the CNL "blank" states), not appearing for AgNP-S, suggesting a different nature between the two phytocompounds. Some considerations should be necessarily given to the reader, being crucial for the purpose depicted by the presented paper. Furthermore a dedicated sentence should be inserted in order to exclude the possibility that this shoulder at lower nm can indicate another populations of particles size.

Line24/25: Do the authors have any hypothesis about nm shifts, between AgNP-L and AgNP-S peaks? Usually in literature UV-Vis bathochromic or hypsochromic shifts are correlated to different NPs dimensions. This consideration can fit in the next size evaluation made by TEM, unveiling different NPs dimensions.

Line28: This statement should contain a correlated reference (see comment at Pag10 Line58)

Figure 5: The AgNPs obtained with 5% extract concentration are, both for L and S, shifted to higher wavelengths. This phenomenon depicts a different size in the material. The opinion of this reviewer is that a short comment should be integrated in the text.

Pag10

Figure 6: Similarly to Figure 5, the same observation can be done for more diluted solutions of AgNO₃ (0.001M).

Line46/50: Similarly to Figure 5 and 6, the same observation can be done for lower T (25°C).

Line58: The reference "50" do not express data about the NPs shape assignation according to registered wavelength. Probably the original insertion was Ref 49, that in any case just cites the useful paper: Optical Properties of Metal Nanoparticles with Arbitrary ShapesJ. Phys. Chem. B 2003, 107, 26, 6269-6275. Please add the correct reference.

Pag13

Line6: The symbol * is not properly reported inside the figure.

Line46: For the sake of clarity, authors should briefly report in the text how they exclude the formation of any traces of silver oxides. This information can be easily extrapolated from the XRD, and is opinion of this referee that this point should be encountered, because of the numerous literature examples of oxides synthetized with similar approaches. A related sentence will enrich and valorise both the novelty and the efficacy of the presented work.

Pag15

Line21: Similarly to Pag 3 line19, the reported NP dimensions should be expressed according to the applied techniques and with the proper. Furthermore, 119 probably is a typo of 129.9 nm.

Reviewer: 2

Comments to the Author(s)

Authors tried to prepare silver nanoparticle with help of natural herb of leaf and stem extract. There are several studies are reported in this area. No novelty in this study.

Reviewer: 3

Comments to the Author(s)

Major review.

In this manuscript, the authors synthesized AgNPs using the aqueous extracts of *C. nutans* leaves and stems through a non-toxic, cost effective and eco-friendly method and determined the classes of phytochemicals, to biosynthesis and characterise the AgNPs. The structure of the manuscript is clear, but there are some grammatical problems in details and some confusion in the selection of the optimal conditions for synthesis. Therefore, I recommend this manuscript to be accepted after solving the following issues.

1. Some sentences in the manuscript have grammatical errors, the sentence in line 40 page 2 should be "AgNP synthesis has become popular", the sentence in line 47 page 2 should be "has become a major interest". The tenses need to be unified in line 8,9,10 page 3. What's more, the use of phrase "AgNPs synthesis" should be uniform throughout the manuscript.
2. There is no reference to this sentence "The biological activities and surface energy of nanoparticles increase with their surface areas." in the first paragraph in Introduction.
3. You mentioned that the use of plant extracts had become a major interest in AgNPs synthesis, what's new about your method compared to other methods using plant extracts to synthesis AgNPs? You may give comparative experiments or the novelty of your method to explain this.
4. The pictures in figure 1 should be reselected for these pictures are not suitable enough for publication.
5. The variable-controlling approach using part 4.5 is inappropriate. The interference of multiple conditions on the result may be ignored if the optimal conditions of a certain factor is selected as the premise in the selection of optimal conditions of next factor. It is suggested to use the orthogonal experiment method, so it is necessary to give the mathematical thought of designing the orthogonal experiments and add supplementary experiments.
6. The X-coordinate and Y-coordinate in UV-VIS spectra in figure 4,5,6,7 are not clear enough.

Author's Response to Decision Letter for (RSOS-200065.R0)

See Appendix A.

RSOS-200065.R1 (Revision)

Review form: Reviewer 3

Is the manuscript scientifically sound in its present form?

Yes

Are the interpretations and conclusions justified by the results?

Yes

Is the language acceptable?

Yes

Do you have any ethical concerns with this paper?

No

Have you any concerns about statistical analyses in this paper?

No

Recommendation?

Accept as is

Comments to the Author(s)

The authors addressed most of my questions in the revised manuscript. Therefore, this manuscript is acceptable to be published in this journal.

Decision letter (RSOS-200065.R1)

Dear Dr Llm:

Title: Optimisation of biogenic synthesis of silver nanoparticles from flavonoid-rich *Clinacanthus nutans* leaf and stem aqueous extracts

Manuscript ID: RSOS-200065.R1

It is a pleasure to accept your manuscript in its current form for publication in Royal Society Open Science. The chemistry content of Royal Society Open Science is published in collaboration with the Royal Society of Chemistry.

RSC Associate Editor: 1
Comments to the Author:
(There are no comments.)

RSC Associate Editor: 2
Comments to the Author:

(There are no comments.)

Reviewer(s)' Comments to Author:

Reviewer: 3

Comments to the Author(s)

The authors addressed most of my questions in the revised manuscript. Therefore, this manuscript is acceptable to be published in this journal.

Appendix A

RSOS revisions

Reviewer 1

Comments	Actions/ amendments
Pag3 Line19: The reported NP dimensions should be expressed according to the applied techniques (in this case DLS) and with the proper form (i.e. 114 nm should be 114.7 ecc...). Furthermore, as correctly said in the text (Pag 12 line 4/5), DLS is not able to discriminate clusters from NPs. A more representative information could be offered reporting the TEM NPs average size (AgNP-L: 101.18 nm, AgNP-S: 75.38 nm).	DLS size corrected accordingly. The average size from TEM was added to the abstract.
Pag4 Line8 and 10 too many repetitions of C. nutans.	Amended
Pag5: Line 35/36: is the L/L ratio (95/5) been optimized before? Line38: Do the authors expect AgNPs to be stable at that temperature under air? Oxidations can be ignored? Line54-55: It is not clear which part of the sample was analysed.	95/5 ratio was optimised. The studied ratios were 95/5, 90/10 and 85/15. Yes, AgNPs may oxidise when exposed in the air. Approximately 20% of AgNPs were reported to oxidise post synthesis (DOI: 10.1039/C5RA10187E). The sample from incubation was screened using UV-Vis spectra to observe the reduction of silver ions to silver nanoparticles. (the raw leaves and stem extract were screened as well)
Pag6 Line47-50: The opinion of this referee is that a better contextualization of the state of the art should be necessary to valorise the work presented.	Sentence revised. (page 6, below table 2)
Pag7 Line25: For the sake of comparison, it should be better to express concentrations with the same dimensions (micrograms or milligrams). Line34-35: It is not completely correct to assign the flavonoids antioxidant activity simply to hydroxyl groups, considering that rings and insaturations are involved in the radical polyphenol stabilization, once the molecule scavenged the oxidation source. Furthermore, even phenolic acids such as compounds belonging to the families of	The data of TPC and TFC have already shown the standard presentation. The contents of phenolic/ flavonoid compounds per 1 mg of extract were reported. In this case, µg of phenolic/ flavonoid content in 1 mg of extract. Statement added.

hydroxycinnamic acids and hydroxybenzoic acids should be cited in the text, considering the role displayed by carboxyl groups, according to the evidences reported in the work (i.e. Pag14 line 60 but above all Pag15 line 10/11). Line 41: Are the solutions always exposed to sunlight? Silver is sensible to light, and the irradiation can play a critical role in the NPs formation. It is opinion of this referee that a comparison test should be conducted in dark conditions, with all the other optimized parameters unchanged (CNL and CNS).	The incubation steps during AgNPs syntheses were conducted in dark environment. The processing step after the incubation was also done in the lowest exposure to light/sunlight.
Pag8 Line40: Please, modify magnetic in electromagnetic.	Corrected
Pag9 General UV-vis: Authors did not make any consideration about the presence of a shoulder for AgNP-L system. This peak seems to be peculiar of the leaves extract (as the CNL “blank” states), not appearing for AgNP-S, suggesting a different nature between the two phytochemicals. Some considerations should be necessarily given to the reader, being crucial for the purpose depicted by the presented paper. Furthermore a dedicated sentence should be inserted in order to exclude the possibility that this shoulder at lower nm can indicate another populations of particles size. Line24/25: Do the authors have any hypothesis about nm shifts, between AgNP-L and AgNP-S peaks? Usually in literature UV-Vis bathochromic or hypsochromic shifts are correlated to different NPs dimensions. This consideration can fit in the next size evaluation made by TEM, unveiling different NPs dimensions. Line28: This statement should contain a correlated reference (see comment at Pag10 Line58) Figure 5: The AgNPs obtained with 5% extract concentration are, both for L and S, shifted to higher wavelengths. This	The presence of the shoulder for AgNP-L and leaves extracts was due to the high flavonoid content. Sentence added. Sentence added. Added in Sosa et al, 2003 Same sentence as comment in line24/25.

phenomenon depicts a different size in the material. The opinion of this reviewer is that a short comment should be integrated in the text.	
Pag10 Figure 6: Similarly to Figure 5, the same observation can be done for more diluted solutions of AgNO₃ (0.001M). Line46/50: Similarly to Figure 5 and 6, the same observation can be done for lower T (25°C). Line58: The reference “50” do not express data about the NPs shape assignation according to registered wavelength. Probably the original insertion was Ref 49, that in any case just cites the useful paper: Optical Properties of Metal Nanoparticles with Arbitrary Shapes J. Phys. Chem. B 2003, 107, 26, 6269-6275. Please add the correct reference.	Sentence added. Sentence added. Added in Sosa et al, 2003
Pag13 Line6: The symbol * is not properly reported inside the figure. Line46: For the sake of clarity, authors should briefly report in the text how they exclude the formation of any traces of silver oxides. This information can be easily extrapolated from the XRD, and is opinion of this referee that this point should be encountered, because of the numerous literature examples of oxides synthetized with similar approaches. A related sentence will enrich and valorise both the novelty and the efficacy of the presented work.	The symbol has been corrected in new figure. The statement regarding silver oxide peaks has been added.
Pag15 Line21: Similarly to Pag 3 line19, the reported NP dimensions should be expressed according to the applied techniques and with the proper. Furthermore, 119 probably is a typo of 129.9 nm.	Amended

Reviewer 3

Comments	Actions/ amendments
1. Some sentences in the manuscript have grammatical errors, the sentence in line 40 page 2 should be “AgNP synthesis has become popular”, the sentence in line 47 page 2 should be “has become a major interest”. The tenses need to be unified in line 8,9,10 page 3. What’s more, the use of phrase “AgNPs synthesis” should be uniform throughout the manuscript.	Amended
2. There is no reference to this sentence “The biological activities and surface energy of nanoparticles increase with their surface areas.” in the first paragraph in Introduction.	Citation has been added.
3. You mentioned that the use of plant extracts had become a major interest in AgNPs synthesis, what’s new about your method compared to other methods using plant extracts to synthesis AgNPs? You may give comparative experiments or the novelty of your method to explain this.	The use of different types of plants as preliminary study.
4. The pictures in figure 1 should be reselected for these pictures are not suitable enough for publication.	Figure has been changed.
5. The variable-controlling approach using part 4.5 is inappropriate. The interference of multiple conditions on the result may be ignored if the optimal conditions of a certain factor is selected as the premise in the selection of optimal conditions of next factor. It is suggested to use the orthogonal experiment method, so it is necessary to give the mathematical thought of designing the orthogonal experiments and add supplementary experiments.	Thanks for the suggestion of using DOE for optimisation. However, we are currently unable to conduct due to budget constraint and we are under Movement Control Order where all labs are closed at the moment due to COVID-19 pandemic. Furthermore, this is a proof-of-concept for the optimisation of efficacy of the nanoparticles. Besides that, we focus on One-Factor-at-Time (OFAT) optimisation method.
6. The X-coordinate and Y-coordinate in UV-VIS spectra in figure 4,5,6,7 are not clear enough	The resolution of the figures has been revised.